

# Malformations in Late Devonian brachiopods from the western Junggar, NW China and their potential causes

Ruiwen Zong and Yiming Gong

State Key Laboratory of Biogeology and Environmental Geology, and School of Earth Sciences, China University of Geosciences, Wuhan, China

## ABSTRACT

Although malformations are found in both extant organisms and the fossil record, they are more rarely reported in the fossil record than in living organisms, and the environmental factors causing the malformations are much more difficult to identify for the fossil record. Two athyrid brachiopod taxa from the Upper Devonian Hongguleleng Formation in western Junggar (Xinjiang, NW China) show distinctive shell malformation. Of 198 *Cleiothyridina* and 405 *Crinisarina* specimens, 18 and 39 individuals were malformed, respectively; an abnormality ratio of nearly 10%. Considering the preservation status and buried environment of the abnormal specimens, and analysis of trace elements and rare earth elements from whole-rock and brachiopod shells, we conclude that the appearance of malformed athyrids is likely related to epi/endoparasites, or less likely the slightly higher content of heavy metal in the sea.

## INTRODUCTION

Abnormalities are common in living organisms; the term usually refers to soft body or skeletal tissue malformation of individuals that occur during ontogeny. However, malformations are known from the fossil record too, and have been reported from individuals of different fossil groups, including foraminifera (*Ballent & Carignano, 2008*), trilobites (*Owen, 1985*; *Babcock, 1993*), brachiopods (*Copper, 1967*; *He et al., 2017*), bivalves (*Savazzi, 1995*), gastropods (*Lindström & Peel, 2010*), cephalopods (*De Baets, Keupp & Klug, 2015*; *Hoffmann & Keupp, 2015*; *Mironenko, 2016*; *De Baets, Hoffmann & Mironenko, 2021*), echinoderms (*Thomka, Malgieri & Brett, 2014*), graptolites (*Han & Chen, 1994*), insects (*Vršanský, Liang & Ren, 2012*), conodonts (*Weddige, 1990*), shark teeth (*Itano, 2013*), amphibians (*Witzmann et al., 2013*), reptiles (*Buffetaut et al., 2007*), primate teeth (*Tougard & Ducrocq, 1999*), and plankton (*Vandenbroucke et al., 2015*; *Bralower & Self-Trail, 2016*). In addition to gene mutations or embryonic developmental disorders, malformed fossils may also have resulted from healed injuries and pathology (*Owen, 1985*; *Babcock, 1993*; *Kelley, Kowalewski & Hansen, 2003*; *Vinn, 2017*, *2018*). Malformed fossils provide important evidence of both organisms-organisms and organisms-environment relationships during geological history. For example, malformed

Corresponding author
Yiming Gong, ymgong@cug.edu.cn

specimens caused by predatory attacks provides us information about the food chain at that time or about the position of prey in the ecological chain (*Kelley, Kowalewski & Hansen, 2003*). Moreover, some malformations resulting from diseases or developmental disorders are likely to be related to the habitat of the organism, such as changes in environmental factors, and parasite, viral or bacterial infection (*e.g.*, *Morris, 1981*; *Rouse, 2005*; *Vandenbroucke et al., 2015*; *De Baets et al., 2021*).

Although many malformed fossils have been described, abnormal fossils (especially macrofossils) are generally rare, with sometimes only one or two specimens known. Therefore, previous studies have generally been limited to description of malformed specimens and simple classification of the cause(s) of malformation, only in a few cases the relationship between malformed specimens and their habitat has been discussed (*Copper, 1967*; *Vandenbroucke et al., 2015*; *Bralower & Self-Trail, 2016*; *He et al., 2017*). Although many malformed fossils are believed to result from developmental disorders or have pathological causes, the environmental factors responsible for the developmental disorders or diseases are mostly indefinite or speculative. The low number of malformed specimens available often limits further study. The Upper Devonian succession in western Junggar, Xinjiang, NW China, contains abundant, well-preserved brachiopods. We collected more than 600 athyrid (*Cleiothyridina* and *Crinisarina*) specimens from the Upper Devonian Hongguleleng Formation in the Buninuer section, of which nearly ten percent of individuals were malformed. The aims of current study are to explore the biotic or abiotic stressors of these malformed brachiopods, and the effect of change in environmental factors on the brachiopod shells.

## MATERIALS AND METHODS

The material studied in this paper was collected from the Upper Devonian Hongguleleng Formation in western Junggar, Xinjiang. The Hongguleleng Formation is a widely distributed marine unit near the Devonian–Carboniferous boundary in western Junggar. The formation is divided into three members: the Lower Member is composed of thin bioclastic limestones, muddy limestones and shales; the Middle Member is mainly made up of fine pyroclastic rocks with a few sandy and muddy limestones; and the Upper Member consists of calcareous clastic rocks with a small amount of bioclastic limestones (*Hou et al., 1993*). The formation is mostly Famennian in age (*Ma et al., 2017*; *Zong et al., 2020*; *Shen et al., 2021*; *Stachacz et al., 2021*). The Hongguleleng Formation is very rich in many types of the early Famennian fossils, such as acritarchs, bivalves, brachiopods, bryozoans, cephalopods, chondrichthyans, conodonts, conulariids, corals, echinoderms, gastropods, ostracods, plants, radiolarians, spores, trace fossils, and trilobites (*Liao, 2002*; Zong R W, 2021, unpublished specimens).

Brachiopods occur in all three members of the Hongguleleng Formation. Brachiopod abundance and diversity is highest in the Lower Member, with the groups present including Productida, Orthida, Rhynchonellida, Athyridida and Spiriferida (*Zong et al., 2016*; *Zong & Ma, 2018*). Athyrids are most abundant in the Lower Member, with only a few athyrid specimen recovered from the base of the Middle Member and the limestone interlayer of the Upper Member (*Zong et al., 2016*). All the athyrids studied in this paper

were extracted from the bioclastic limestone in the upper part of the Lower Member of the Hongguleleng Formation in the Buninuer section, 15 km north of Hoxtolgay town. This section is located about 14 km southwest of the Bulongguoer section, which is the type section of the Hongguleleng Formation (*Hou et al., 1993*). The lithology and fossil assemblages of the Buninuer section are the same as those of the stratotype section, the upper part of the Lower Member of the Hongguleleng Formation of both sections were deposited in a distal storm lithofacies sedimentary environment (*Fan & Gong, 2016*). A total of 603 athyrids in two genera (*Crinisarina* and *Cleiothyridina*) are non-flattened specimens with well-preserved dorsal and ventral valves. Although athyrids occur in other beds of the Hongguleleng Formation, no malformed specimens were found in those levels.

The 603 specimens include a wide range of size and may include individuals representing different growth stages (File S1). We divided the shell length (L) into six size classes: 5 mm ≤ L < 10 mm; 10 mm ≤ L < 15 mm; 15 mm ≤ L < 20 mm; 20 mm ≤ L < 25 mm; 25 mm ≤ L < 30 mm and 30 mm ≤ L < 35 mm, and counted the number of malformed specimens in each class. The length of all athyrids were measured by a vernier caliper. All photographs were taken using a Nikon D5100 camera with a Micro-Nikkor 55 mm f3.5 lens.

To explore whether athyrid malformations were caused by environmental factors, trace and rare earth elements of whole-rock samples from specific levels within the Hongguleleng Formation were measured. Samples BL–1 and BL–2 were obtained from the lower part of the Lower Member of the Hongguleleng Formation, which yielded abundant non-malformed athyrids. Samples BL–3 and BL–4 came from the upper part of the Lower Member, form where the malformed fossils described in this paper were obtained. Samples BL–5, BL–6 and BL–7 were collected from the Middle Member of the Hongguleleng Formation; only a few athyrids occurred at the bottom of this member, and the group almost disappeared above that level. Sample B9b–1 was from the Upper Member, which yielded a small number of athyrids. The trace elements and rare earth elements of four athyrid shells were measured. Samples CL69 and CR178 are non-malformed shells, while CLJ13 and CRJ29 are malformed shells. All samples were ground into powder and analyzed by ALS Minerals/ALS Chemex Co. Ltd (Guangzhou, China). Rare earth and trace elements were fused with lithium borate, and quantitatively analyzed by ICP-MS with Elan 9000 Perkin Elmer that was made in America. The $Ce_{anom}$ is equal to $lg[3Ce_n/(2La_n + Nd_n)]$, and $Ce_n$, $La_n$ and $Nd_n$ were NASC-normalized of Ce, La and Nd, respectively.

## RESULTS

Of the 603 athyrid fossils, macroscopic abnormalities were detected in 57 specimens. The most common teratomorphy is obvious asymmetry on the left and right sides of the shells (Figs. 1B–1E, 1G–1J), significantly different from common, non-malformed specimens (Figs. 1A, 1F). Malformation is more obvious on the dorsal valves, and is mainly visible as significantly widening or narrowing on one side of the shell (Figs. 1D1, 1G1, 1H1, 1J1). Near the anterior border of the dorsal valves, the grooves on both sides of the fold are significantly different from those of non-malformed specimens, with some grooves being wider (Figs. 1B1, 1C1, 1E1, 1I1), others being narrower and some almost

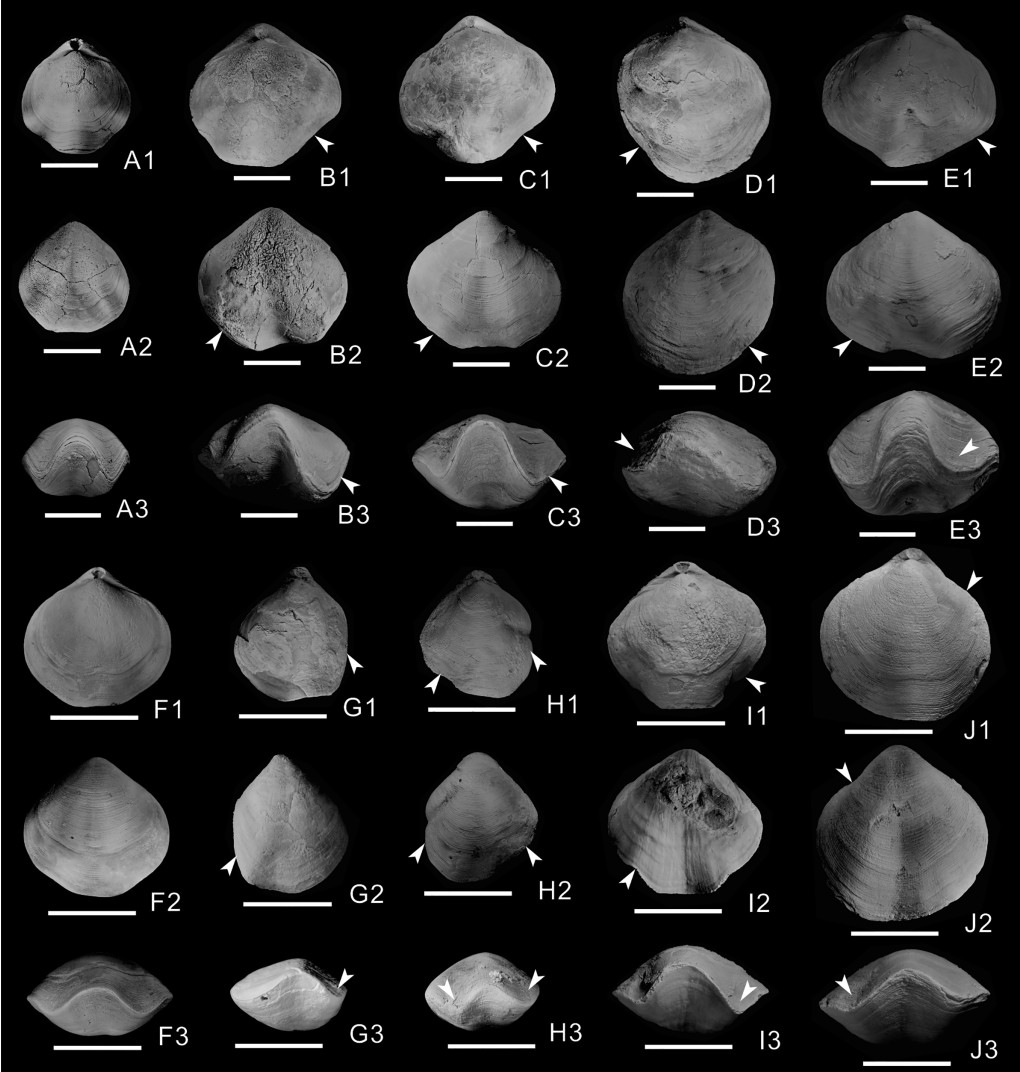

**Figure 1 Athyrids from the upper part of the Lower Member of the Hongguleleng Formation (Upper Devonian) in western Junggar.** (A–E) *Crinisarina*: (A) (specimen number BGEG-CR324) is a non-malformed specimen; (B–E) (specimen numbers BGEG-CRJ05, BGEG-CRJ10, BGEG-CRJ18 and BGEG-CRJ17) are malformed specimens, with malformations indicated by white arrows. (F–J) *Cleiothyridina*, (F) (specimen number BGEG-CL98) is a non-malformed specimen; (G–J) (specimen numbers BGEG-CLJ06, BGEG-CLJ01, BGEG-CLJ15 and BGEG-CLJ13) are malformed specimens, with malformations indicated by white arrows. All scales are 10 mm.

disappearing (Figs. 1D1, 1G1). On the ventral valves, in addition to the unequal size on either side of the shell, the sulcus is slightly curved in some malformed specimens (Figs. 1B2, 1E2, 1I2). In frontal view, the asymmetry is more obvious, and is mainly manifested as different depths and widths of the grooves on both sides of the fold (Figs. 1B3–1E3, 1G3–1J3); for example, the grooves on one side of some specimens become deeper and wider, up to twice as much as those on the non-malformed side (Fig. 1B3). In addition, the grooves of some specimens become shallower and narrower (Fig. 1H3), even almost disappearing on one side of a few specimens (Figs. 1D3, 1G3). In the

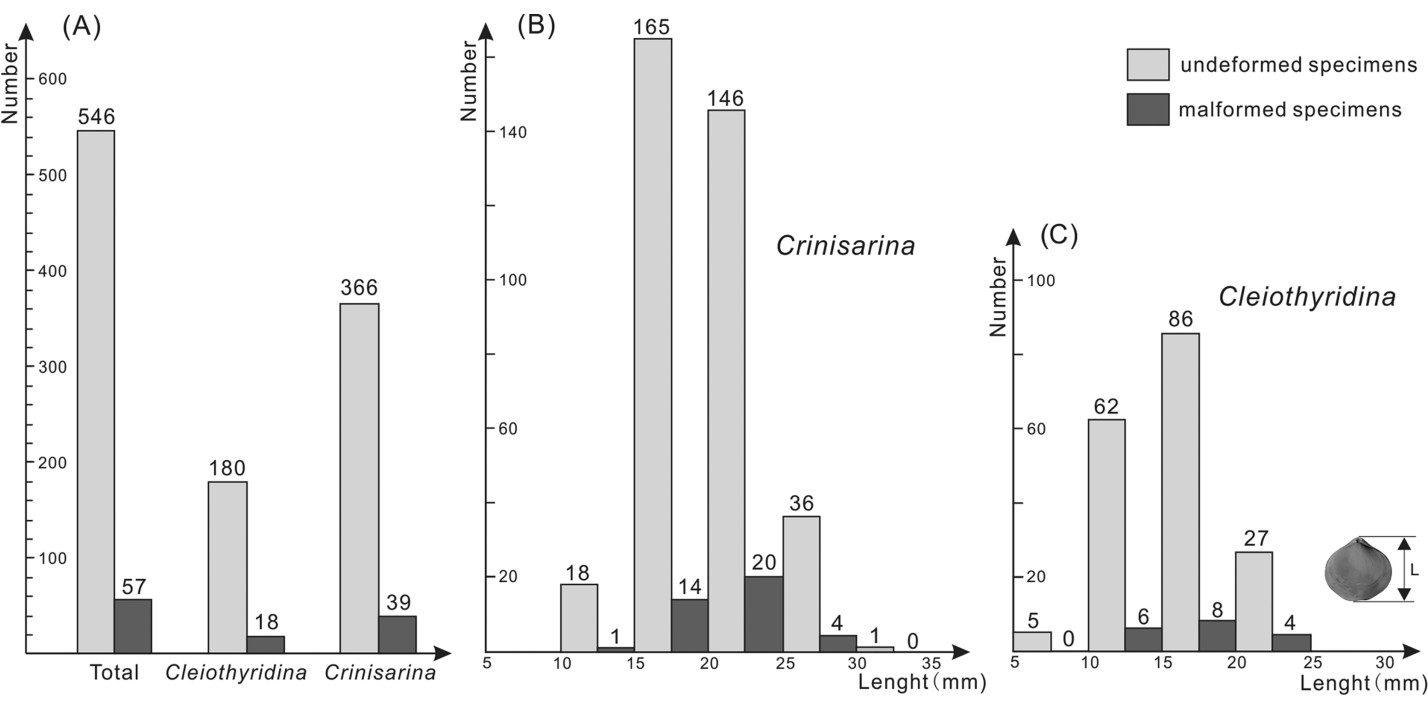

**Figure 2** Histogram showing the number of malformed athyrids (A) and the distribution of non-malformed and malformed specimens in different size classes (B–C) from the Upper Devonian Hongguleleng Formation in western Junggar.

malformed specimens, the commissure on the front of the dorsal and ventral valves forms irregular wavy lines, markedly different from the regular wavy lines in non-malformed specimens (Figs. 1A3, 1F3).

Of the 198 specimens of *Cleiothyridina* and 405 of *Crinisarina*, 18 (9.1%) *Cleiothyridina* and 39 (9.63%) *Crinisarina* were malformed. The overall malformation ratio was 9.45%, nearly one-tenth of all specimens (Fig. 2A). In all malformed specimens, the distribution of malformations is asymmetric on the shells in dorsal view, malformations occur in the right side of 25 shells of *Crinisarina*, but there are only 14 in the left side of shells. For the *Cleiothyridina*, malformation occur in the right side of 14 shells, while in the left side of three shells, and in both sides of one shell (File S1). Moreover, malformed individuals occur in almost all size classes (Figs. 2B, 2C). The malformation percentages of *Crinisarina* are 5.56% (10 mm ≤ H < 15 mm), 8.5% (15 mm ≤ H < 20 mm), 13.7% (20 mm ≤ H < 25 mm) and 11.1% (25 mm ≤ H < 30 mm); those of *Cleiothyridina* are 9.68% (10 mm ≤ H < 15 mm), 9.3% (20 mm ≤ H < 25 mm) and 14.8% (25 mm ≤ H < 30 mm). Thus, shell-malformation occurs at different athyrid growth stages, and the probability of abnormality is higher in larger specimens. That indicates a higher probability of malformation during advanced ontogenetic stages of the studied brachiopod taxa.

## DISCUSSION

Western Junggar is part of the Central Asian Orogenic Belt (*Buckman & Aitchison, 2004*; *Windley et al., 2007*). This region experienced strong tectonic activity during the

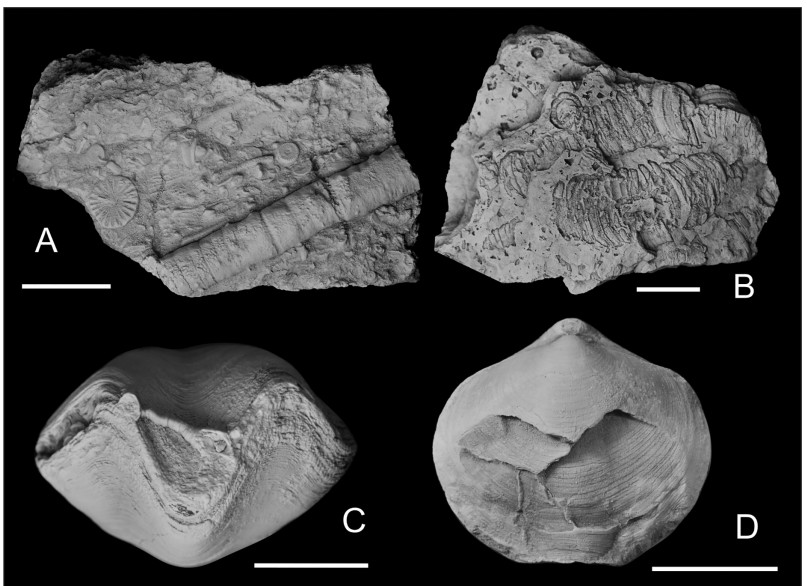

**Figure 3 Some crinoid stems and athyrids from western Junggar.** (A) Three-dimensional crinoid stem preserved in muddy limestone from the upper part of the Lower Member of the Hongguleleng Formation in the Buninuer section; (B) flattened crinoid stem preserved in the calcareous siltstone distorted by tectonic activity, Carboniferous Hala'alate Formation, western Junggar; (C) coral parasitizing a non-malformed shell of *Crinisarina* (specimen number BGEG-CR44) from the Lower Member of the Hongguleleng Formation; (D) *Cleiothyridina* (specimen number BGEG-CL56) cracked before lithification of the sediment, obviously different from the malformed specimens. All scales are 10 mm.

Paleozoic, resulting in different degrees of metamorphism or deformation of the Paleozoic strata in the study area (*Xu et al., 2009*; *Gong & Zong, 2015*; *Wang & Zhang, 2019*). Athyrids exhibiting left-right asymmetry might have resulted from tectonic deformation; however, there is no obvious stratal deformation in the Hongguleleng Formation in the Buninuer section. This section has yielded fossils (*e.g.*, trilobites, crinoids, and corals) that are well-preserved in three dimensions (Fig. 3A), which obviously differs from specimens obtained from distorted strata affected by tectonic deformation (Fig. 3B). Moreover, asymmetry was not detected in other brachiopods from the same layer, so the malformed specimens were not affected by the tectonic activity. A very small number of athyrid specimens cracked before lithification of the sediment are also significantly different from these asymmetric specimens, and they can be easily distinguished (Fig. 3D).

Epibionts can cause malformation of their hosts; such phenomena have also been reported for shelly fossils (*e.g.*, *Klug & Korn, 2001*; *Checa, Okamoto & Keupp, 2002*; *Zatoń & Borszcz, 2013*; *Mironenko, 2016*; *Stilkerich, Smrecak & De Baets, 2017*). Some athyrids from the Upper Devonian of western Junggar bear epibionts, such as corals and bryozoans. However, epibionts are not found on any malformed specimens; on the contrary, specimens bearing shelled epibionts or epibiont with mineralized shells/skeletons are all non-malformed shells (Fig. 3C). Therefore, it is unlikely that these athyrid teratomorphies were caused by epibionts. However, endoparasitic organisms cannot be ruled out as a teratogenic factor (*e.g.*, *Savazzi, 1995*; *Vinn, Wilson & Toom, 2014*), as well as
epiparasitic shell-less (soft-bodied) organisms and microorganisms, they are likely to cause malformations in these athyrid shells from western Junggar. For the specific identity of endoparasites, shell-less organisms and microorganisms, it is difficult to confirm because of their poor preservation or they are indiscernible in the fossil record. Some brachiopod malformations were caused by predators, which presented the fractures, indentations, and scars on their shells, and were often accompanied by signs of repair (*e.g.*, *Alexander, 1986*; *Kowalewski, Flessa & Marcot, 1997*; *Happer, 2005*; *Vinn, 2017*). In these malformed athyrid shells from western Junggar, except for a pair of indentations on the opposite valves of specimen BGEG-CLJ01 (Fig. 1H), no wounds or scars were found on other specimens. Most specimens only showed left-right asymmetry of the shells, reflecting that predation is not the main cause of the malformation, but predators may have preyed on the malformed athyrids.

Malformations of organisms may also be related to their living environment. Changes in certain environmental factors, such as oxygen deficiency or excessive organic matter, heavy metals, and toxic elements, often lead to the malformation or even death of organisms. The high number of malformed athyrids from the Upper Devonian strata might be related to the marine environment in western Junggar at that time. Excessive organic matter is a common factor, and eutrophication has been identified in the Late Devonian sea (*Murphy, Sageman & Hollander, 2000*). *Suttner et al. (2014)* found that there were no significant changes in the total organic carbon (TOC) content through the Lower Member of the Hongguleleng Formation at its type locality, *i.e.*, the TOC content of the beds with malformed specimens was basically the same as that of sediments with only non-malformed specimens. *Copper (1967)* studied abnormalities of the Devonian brachiopod *Kerpina* in the Eifel region, Germany, and concluded that the variations in the shell-morphology resulted from the influence of bottom currents on the immobile *Kerpina*, which had a thick, short pedicle. However, this mechanism cannot be used to explain the malformation of these athyrids in western Junggar, because large numbers of other benthic organisms (*i.e.*, brachiopods, corals, bryozoans and stromatoporoids), which are all non-malformed in the same layer. In addition, the fossils preserved in the upper part of the Lower Member of the Hongguleleng Formation are relatively complete, and there is no evidence of strong bottom currents, so the influence of bottom currents can be excluded as a teratogenic factor. *Hoel (2011)* found the shells of brachiopod *Pentlandina loveni*, from the Sliurian Högklint Formation in Gotland (Sweden), are commonly markedly asymmetric, and some groups of shells occur in tight clusters, each apparently attached to other shells of the same species. He interpreted that these asymmetrical shells resulted from the limited space for growth, *i.e.*, overcrowded conditions. However, all specimens from western Junggar are isolated, instead of grouped in tight clusters or attached to other shells. Furthermore, if they are living in overcrowded space, the distribution of malformation should be random or almost uniform on both sides of the shells, but the malformations commonly occur on the right side of athyrids shells (dorsal view) from western Junggar (File S1), so the overcrowded conditions can be excluded.

Marine hypoxia could also lead to brachiopod malformations. *He et al. (2017)* for example, proposed that hypoxia was a major factor in the miniaturization of brachiopods

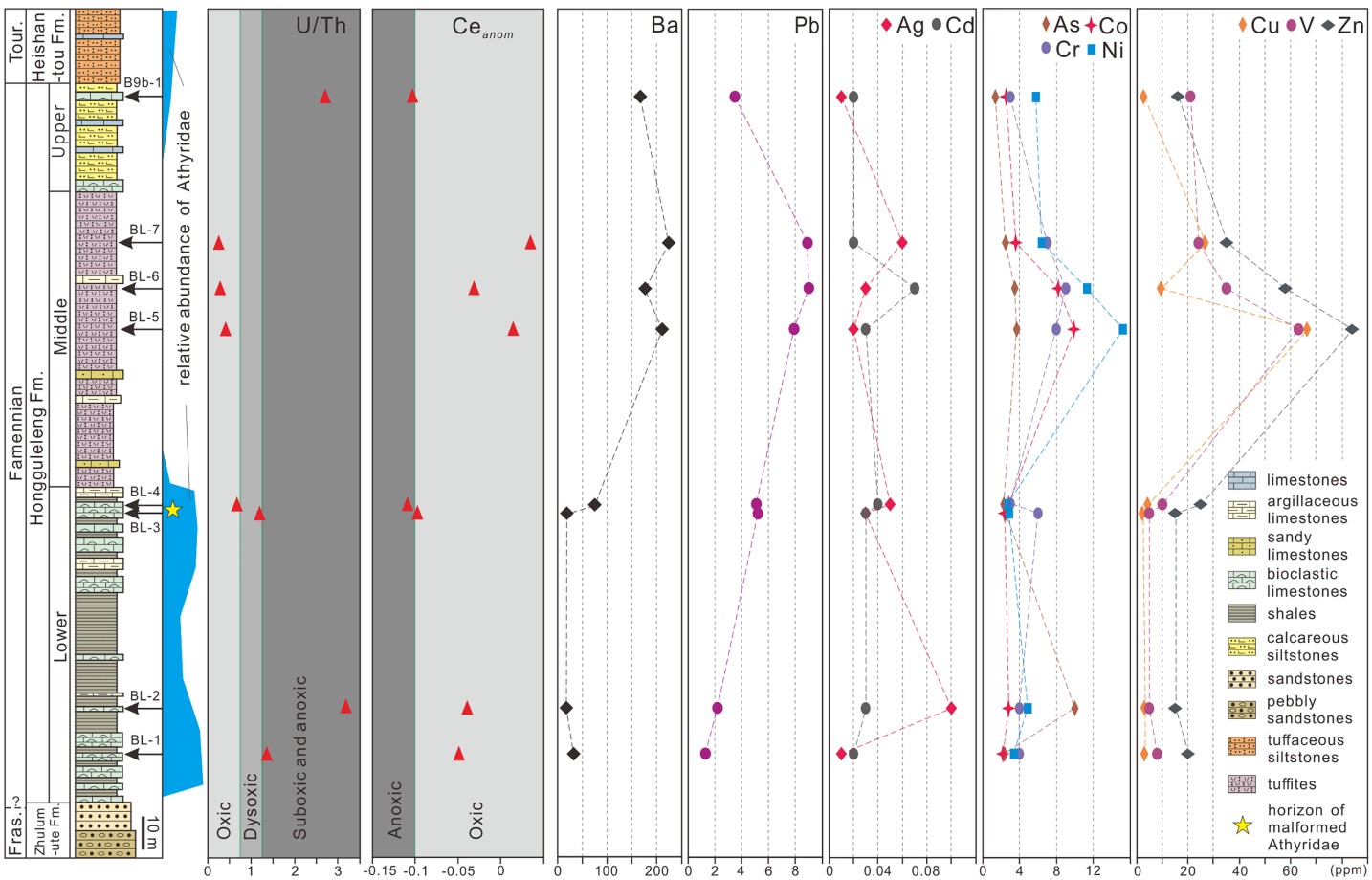

**Figure 4** U/Th ratio, Ce$_{anom}$ and distribution of heavy metals and toxic elements in whole rocks in the Upper Devonian Hongguleleng Formation in western Junggar.

during the end-Permian in southern China. U/Th and Ce$_{anom}$ are often used as indicators of marine hypoxia (*Jones & Manning, 1994*; *Carmichael et al., 2014, 2016*). For the sediments of the Hongguleleng Formation at the Buninuer section, the U/Th ratio of sample BL–4 fell into the oxic range, whereas that of sample BL–3 fell into the dysoxic range; both Ce$_{anom}$ values were near the oxic–anoxic boundary (Fig. 4, File S2). However, the uranium enrichment factor (U$_{EF}$, *Algeo & Liu, 2020*) of the above two samples reached 0.69 and 0.19 (in comparison to average limestones, *Wedepohl, 1970*), *i.e.*, the uranium is depleted, indicating an oxic condition. In addition, a low TOC content (*Suttner et al., 2014*), shallow-marine benthic fossils (e.g., corals, trilobites, brachiopods and echinoderms) occuring in abundance in the same horizon together with malformed athyrids (*Liao, 2002*), rich trace fossils reflected in a not oxygen restricted benthic environment (*Fan & Gong, 2016*; *Stachacz et al., 2021*), and the beds lack sedimentary indicators of anoxia (such as black shale), also suggest that hypoxia presumably did not occur during deposition of the upper part of the Lower Member of the Hongguleleng Formation.

High levels of heavy metals or toxic elements can also lead to malformation of organisms' soft and hard tissue, as has been proven for a large number of living organisms (*Wang, Yang & Wang, 2009*; *Ma et al., 2011*; *Zhao et al., 2017*; *Lasota et al., 2018*; *Riani, Cordova & Arifin, 2018*). For example, when copper (Cu) and zinc (Zn) were added to the water for feeding the foraminiferan *Ammonia beccarii*, the organisms developed abnormalities (*Sharifi, Croudace & Austin, 1991*), and when scallops were placed in wastewater from a gold mine with concentrations of 14% and 50% for 6 h, the abnormality ratio increased by 6% and 21%, respectively (*Ma et al., 2011*). Sediments have been demonstrated to be an important source of heavy metals for benthic animals (*Wang, Stupakoff & Fisher, 1999*). The levels of heavy metals and toxic elements through the Hongguleleng Formation are presented in Fig. 4 and File S2. The levels of some heavy metals (*e.g.*, cadmium, lead, barium and zinc) in the layer with the malformed athyrid shells are relatively high compared to the levels in the lower part of the Lower Member, particularly cadmium and lead. The levels of cadmium are 0.02 and 0.03 ppm in the lower part of the Lower Member, and are 0.03 and 0.04 ppm in the horizon that yielded the malformed fossils. The lead contents are 1.3 and 2.2 ppm in the lower part, but 5.1 and 5.2 ppm in the upper part. In the Middle Member of the Hongguleleng Formation, where athyrids almost disappeared, the heavy metal levels are even higher. In the Upper Member, where athyrids reappear, the heavy-metal content decreases again (Fig. 4). Thus, the abundance of athyrids is negatively correlated with the levels of heavy metals, but the number of malformed specimens is positively correlated with that, especially that of lead (Fig. 4). Furthermore, the levels of most heavy metals and toxic elements are slightly higher in malformed shells than in non-malformed shells, particularly lead, silver, cobalt and arsenic (File S3). In the studied specimens of teratomorphic *Cleiothyridina* and *Crinisarina*, the lead, silver, cobalt and arsenic levels were all higher than those of non-malformed shells. In *Cleiothyridina* the levels of lead, silver, cobalt and arsenic were 2.3, 0.01, 2.0 and 2.9 ppm, respectively, in non-malformed shells, but 2.6, 0.03, 2.5 and 3.6 ppm, respectively, in malformed shells. In *Crinisarina*, the levels of lead, silver, cobalt and arsenic in non-malformed shells were 1.9, <0.01, 2.6 and 2.2 ppm, respectively, whereas in the malformed specimens, the levels were 7.2, 0.01, 3.2 and 2.5 ppm, respectively (File S3). However, the contents of cobalt and other heavy metals (*i.e.*, copper, chromium and vanadium) did not change significantly in the sediments from the non-malformed athyrids-bearing layers to the layers containing malformed specimens, and the contents of arsenic and silver even decreased in the layers containing malformed athyrids (Fig. 4). Therefore, these heavy metals or toxic elements in greater concentrations may not be related to the malformation of athyrids. Only the lead content in sediments is positively correlated with the athyrid shell malformation, but negatively correlated with the abundance, while the lead content of malformed shells is higher than that of non-malformed shells.

Lead is a common type of marine heavy-metal, and excessive lead content in the sea often leads to malformation or even death of shellfish, or at least affects their growth (*Li, Sun & Li, 2011*). In Upper Devonian strata of western Junggar, the malformation ratio of athyrids is nearly one in ten. In addition, the heavy metal (especially lead) content in

sediments is higher than that of sediments containing only non-malformed shells, and the levels of some heavy metals are higher in malformed shells than in non-malformed shells. However, the difference of lead concent in the shells and surrounding rocks of non-malformed and malformed brachiopods is still within the same order of magnitude, and no malformations have been found in other brachiopods taxa except these two genera. Therefore, it is not entirely certain that slightly higher lead content could cause malformations in these athyrid shells. In the future, geochemical analysis of more athyrid shells with novel state-of-the art methods and a comprehensive comparison of heavy metal content from adjacent horizons and other brachiopods may be able to provide further testing of the link between these malformations and environmental factors.

## CONCLUSIONS

Some specimens of two athyrid genera, *Cleiothyridina* and *Crinisarina*, from the Upper Devonian Hongguleleng Formation in western Junggar are obviously malformed, mainly in the form of asymmetry of the left and right sides of the shells. The abnormality ratio is nearly 10% of specimens. Malformation is apparent in individuals of different sizes, with larger individuals being more likely to exhibit malformation. Based on the study of the burial state and preserved environment of the fossils, and geochemical analysis of the sediments and athyrid shells and comparison with rock material from horizons that did not contain teratomorphic specimens, we hypothesize that the malformations were possibly caused by unidentified (or soft-bodied) epi/endoparasites, or a low probability that slightly high heavy-metal (specifically lead) in the sea, rather than eutrophication, bottom current activity, overcrowded conditions, hypoxia or other factors.

## ACKNOWLEDGEMENTS

We would like to thank Zhen Shen, Chao Guo and Junyan Dong, all from China University of Geosciences (Wuhan) for their help in the field work. We appreciate much the constructive and critical comments from Thomas Suttner, Uwe Balthasar, Michał Rakociński, two anonymous reviewers and Editor Kenneth De Baets, which aided in the further improvement of the manuscript.

### Funding

This work was supported by the National Natural Science Foundation of China (Nos. 42072041, 41702006, 41872034). The funders had no role in study design, data collection and analysis, decision to publish, or preparation of the manuscript.

### Grant Disclosures

The following grant information was disclosed by the authors:
National Natural Science Foundation of China: 42072041, 41702006, 41872034.

### Competing Interests

The authors declare that they have no competing interests.

## Author Contributions

- Ruiwen Zong conceived and designed the experiments, performed the experiments, analyzed the data, prepared figures and/or tables, authored or reviewed drafts of the paper, and approved the final draft.

- Yiming Gong conceived and designed the experiments, performed the experiments, analyzed the data, prepared figures and/or tables, authored or reviewed drafts of the paper, and approved the final draft.

## Data Availability

The raw measurements are available in the Supplemental Files.

The specimens are stored in the State Key Laboratory of Biogeology and Environmental Geology, China University of Geosciences, Wuhan, China: BGEG-CL01 to BGEG-CL180, BGEG-CR01 to BGEG-CR366, BGEG-CLJ01 to BGEG-CLJ18, and BGEG-CRJ01 to BGEG-CRJ39.

## Supplemental Information

Supplemental information for this article can be found online at http://dx.doi.org/10.7717/peerj.13447#supplemental-information.

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
