# Peer review of "Malformations in Late Devonian brachiopods from the western Junggar, NW China and their potential causes"

_PeerJ, doi:10.7717/peerj.13447_

## Round 0.1 · original submission · Major Revisions

Dear authors,

I thank you for your patience in awaiting my decision which was not straightforward given the disparate opinions of the reviewers. You provide interesting new data on malformations in Devonian brachiopods and their potential link with heavy metal concentrations. I feel both the information on malformations (reviewers 1 and 4) as well as trace element data (reviewer 2 and 3) is worth publishing but there are some crucial issues which need to be resolved before publication. I would advise to focus the manuscript on documenting the malformations in your specimens and only subsequently try to link them with potential causes. One cannot rule out the influence of heavy-metals in these malformations but at the moment your arguments are not particular strong (some heavy metal data presented even runs counter to your interpretations). The manuscript/message would be more convincing if data from the brachiopod shells would be available and align with those of the sediments or if at least the highest degree of heavy metal concentrations would correspond with the deformations. It is also not clear to me why particular proxies are highlighted, while other are not (see also reviewer 2).

Please address the following points in the revision:

1) Title: Please use a more general title focusing on the malformations. Something along the lines: “Malformations in the Late Devonian brachiopods from western Junggar, NW China and their potential causes” or “Malformations in Late Devonian brachiopods from the western Junggar, NW China and their potential relationship with heavy metal poisoning”

2) Research questions: Please state more clearly your research questions and hypotheses (see reviewer 1).

3) Documentation of the malformations: you need to more clearly define your what you consider a malformation and what criteria you used to identify them. I agree with reviewer 4 that some types of damage rather reflect predation and linking anomalies with heavy metal poisoning. Presenting the characteristic of malformations of modern brachiopods (or other bivalved shell secreting invertebrates) and they can be tied to abiotic and biotic factors would be crucial for further interpretations (compare also reviewer 2).

4) Potential causes: shell deviations/irregularities can develop in many ways and you exclude many of them prematurely (compare reviewer 2). Unless you have ways to characteristically tie the kind of malformation to particular abiotic or biotic stressors (and direct evidence for that stressors in individual shells rather than surrounding sediments), a more balanced discussion of various possibilities is crucial. Also, better proxies are available and could be used (e.g., redox conditions: see reviewers 2 and 4). I agree with reviewers 2 and 4 that predation and soft-bodied parasites/epizoa could also explain some of the malformations you are here depicting.

5) Relationship with heavy metal poisoning: The influence of heavy metal poisoning seems like an attractive explanation at first but challenging to prove (compare reviewer 4). Also, your data, which is sound, is not entirely consistent with your interpretations. For example: Why also high concentrations of Pb in levels which lack malformed brachiopods or no malformations in levels with high concentrations of Arsenic (compare reviewer 2). A more careful discussion of the correspondence with abiotic proxies (heavy metals, redox/oxygenation) and potential reasons discrepancies would be necessary.

6) In your revisions you should more equally discuss the advantages (pros) and disadvantages (cons) of the heavy metal explanation versus other possible explanations. I see no issue to admit future analyses are necessary to support one particular hypothesis. You need to more clearly highlight which interpretations are more speculative and/or require additional study. This would make your manuscript message clearer and stronger. Reviewer 3 also stressed that some of the previously published geochemical data from Hongguleleng is missing and would be worthwhile to add.

7) Reviewer 1 has suggested that you cite specific references. You are welcome to add it/them if you believe they are relevant. However, you are not required to include these citations, and if you do not include them, this will not influence my decision.

Please address these as well as additional points raised by the reviewers. I feel that when focusing more on the malformations and more equally discussing potential causes, your manuscript would be stronger and easier to follow. I look forward to seeing your revised manuscript.

Reviewer 1 ·

Basic reporting

This is interesting and important paper on heavy-metal-induced malformations in the Late Devonian brachiopods from western Junggar, NW China

The presented material has not been published previously, either in part or in full. The text is properly organized and presented and all objectives and conclusions are clear.
I did not find any errors of facts. I found that all interpretations are competent and sufficiently backed with available data. The list of references is OK.
I found that the length of the manuscript is appropriate. I don’t think that there are any sections that could be shortened or, to the contrary, need to be expanded for clarity.
The figures are necessary, and no additional ones are needed. All illustrative materials are of good quality and adequately explained in the text.

Experimental design

The research questions would be more clearly stated such as "The aims of current study are: "
Methods are described with sufficient information and are reproducible by another investigator.

Validity of the findings

The findings are well presented and I don't have doubt in their validity. The data on which the conclusions are based are provided.

Additional comments

Can be published after minor/moderate revision.

Annotated reviews are not available for download in order to protect the identity of reviewers who chose to remain anonymous.

Reviewer 2 ·

Basic reporting

The paper is nicely prepared and well-written (although I am not a native speaker). It is rally an interesting topic which fit to the journal very well. The rest of the comments are below.

Experimental design

The authors retrieved many geochemical data which potentially may be used for explanation of the problem they want to resolve. The supplemental data containing the elements and their quantity is also provided.

Validity of the findings

The paper submitted by Ruiwen Zong and Yiming Gong concerns an interesting aspects of the occurrence of malformed brachiopods. The authors link the origin of malformations with hostile palaeoenvironment caused by, as the authors stated, toxicity of heavy metals, especially by lead (Pb).
Although the toxicity of the palaeoenvironment is possible, its link to the occurrence of malformations in the brachiopods' shells should be well-supported. The authors indeed found that the high content of Pb correlates with the occurrence of the malformed brachiopods, and the highest Pb content falls within the interval when the brachiopods disappeared. Ok, but some questions arise.
1. Quite high Pb content also occur in the interval when the brachiopods recover, but there is no signs of malformation. Why? And why the whole population of the bachiopods do not bear malformations within this specific horizon? Please explain.
2. Very high content of another very toxic element, arsenic (As), occur below the horizon with malformed brachiopods where actually the brachiopods are very abundant and where the Pb is on much lower level. In the horizon with malformed specimens, As strongly drops. Thus, the authors should explain such discrepancy. Does arsenic cause such malformations, or not? It should be explained.
3. The authors said: "Some athyrids from the Upper Devonian of western Junggar bear parasitic organisms, such as trumpet corals and bryozoans". However, I think these epibionts were not parasites. If so, please provide the evidence.
4. The authors also stated that malformations were unlikely caused by parasites as no epibionts are preserved on the malformed specimens. However, many parasites which could caused shell malformations could be soft-bodied organisms which either lived on or inside the host brachiopods. In that way, such epi/endobionts would not fossilize.
5. Redox proxies. Ni/Co and V/V+Ni are bad redoks proxy. V/Cr is also discussed as not reliable. One of the best I think is U/Th. Vanadium and Nickel proxy often show anoxia in oxic paleoenvironments, and the Ni/Co shows the opposite. So, I would not rely the interpretations solely on these two proxies. If You can, please check it using U/Th, especially that You have the data. Please try and check how their relate to the other proxies used.

Additional comments

Summarizing, this is interesting paper and suitable for the journal. I think many readers would be interested in the problem. However, to strengthen the conclusions, the authors should look at some specific points presented by me above to avoid any oversimplifications and overinterpretations. I would use the phrases like "possible link with toxicity" etc. That's a shame that authors did not analyse the section in a more detail. Now the data are rather very scattered throughout the section. Anyway, having even these which are presented, the authors may retrieve conclusions in elegant, but careful, manner.
For me the paper needs a MODERATE/MAJOR revision before it is published.
Best wishes.

Annotated reviews are not available for download in order to protect the identity of reviewers who chose to remain anonymous.

·

Basic reporting

see: 4. Additional comments

Experimental design

no comment

Validity of the findings

no comment

Additional comments

I have added some few (minor) suggestions and comments for improving the style of the manuscript text directly into the original word document. The authors should go through that and suggest whether these comments can be helpful for them.
The same accounts to my suggestions concerning additional references which for the better presentation of the research carried out at the Hongguleleng should be included. - However, finally it is upon the authors to decide! In case the authors do need some of the suggested literature, please feel free to contact me via: suttner.thomas@gmail.com.

The recognition of malformed brachiopods is a very important observation which has not been recognized for fossils from the Hongguleleng before! - I have experience in seeing similar malformation in Middle Devonian conodonts - but triggers and causes are further speculative. - The presented study by Zong and Gong could be one possible explanation.

·

Basic reporting

The manuscript is generally written in clear professional English. In few places it would benefit from minor corrections, but nothing that impacts on the understanding of the science. The introduction and background is OK for some parts, but lacks information on the regional / local geology. Particularly info on the depositional environment is needed to help the interpretation of the findings. The figures are OK, but Figure 2 could be simplified (the pie charts are not needed and the histograms could be clearer). Figure 4 shows a lot of geochemical data of which some are not discussed in the text (e.g. Ba, Ag, Cd etc.) so these could be removed. Raw data are shared in the form of spreadsheets with geochemical values and brachiopod measurements and listings of malformations. However, see below for a critique of why this is insufficient.

Experimental design

As far as I understand from the manuscript, lead was measured from bulk rock samples, so it remains unresolved where the lead sits within the rock. If an argument is made about the concentrations of lead causing malformations, it is necessary to demonstrate that the lead was in the water – many volcanic minerals will contain some lead, but remain inert in the sediment. If there was significant lead in the water, it would accumulate in the shells of the brachiopods. So these need to be analysed. 5ppm lead from bulk rock probably means that a fraction of this was in the water. But even if 5ppm would have been in the water it would have been too little to account for the reported percentages of malformations.

Validity of the findings

Malformations of Devonian shells induced by lead poisoning is a very challenging hypothesis to assess and needs to convince the reader of: (1) malformations – how are they defined? (2) Existing malformations cannot be explained by expected processes such as space constraints or repair of predation damage; (3) that the lead was (3a) in the water (and not in sedimentary grains) and (3b) was in sufficiently high concentrations. Unfortunately, the manuscript falls short of many of these critical points.

Malformations: While the manuscript qualitatively describes what is considered as malformations, no precise definition is being provided. To what degree does the shell have to deviate from the ‘perfect’ shape to be considered a malformation? What is the natural variability of the shell shape? Shells were binned by size, and the measurement was given as ‘height’. Figure 2C indicates that the author’s ‘height’ is really the length (as defined in volume 1 of the Brachiopod-themed volumes of Treatise on Invertebrate Paleontology). In the absence of a precise definition of what constitutes a malformation and what is considered a healthy shell, it is not possible for the reader to fully evaluate the results. It is also rather difficult to explain away the apparent absence of ‘malformations’ from other groups of fossils in the same strata. Even if different taxa have different degrees of sensitivity – serious Pb concentrations in the environment should leave their mark in the total fauna.

Alternative explanations of the ‘malformations’. Some of the possible alternative causes of the malformations are brushed away too easily. The slight deviation of growth caused by space constraints are discounted because the specimens don’t occur closely together. However, no data is provided about the depositional environment. Where the specimens transported? How do the authors know that they did not grow in a constrained space? Furthermore, repair of predation-induced damage is not discussed at all, when at least one of the figured specimens looks like it suffered that type of damage (Fig. 1H). The ‘tectonic’ influence on the specimens is discussed in too much detail – why not list some papers on the tectonic structures of the region? As long as there is no cleavage / strain present all is fine. The image in Fig. 3D is not an example of tectonic damage – this is just a shell that cracked before lithification of the sediment.

Additional comments

The brachiopods are well documented and I do not doubt that there are malformations in these shells. But I think that you are trying to force the issue with Pb without having evidence. The Devonian biodiversity crisis is a very exciting time when significant paleoecological restructuring is happening. I suspect that a lot of your shell damage is down to predation, which is rarely reported (see for example: Alexander 1986, Journal of Paleontology, v. 60(2), 273-285) and can tell you something about paleoecological relationships. Identifying and quantifying predatory shell damage (and asking why some taxa are more strongly affected than others) might be a more interesting way to evaluate your fossils.

---

## Round 0.2 · Major Revisions

The manuscript has been substantially improved by including discussion on parasites/epizoans, more information on the geological discussion and a more balanced discussion of the potential influence of metal poisoning. I remain of the opinion that I would like to see data and hypothesis published but there are some remaining points which need to be addressed before publication. The main points which need to be addressed are:


1. Background values of high metals in the sampled settings: its needs to be discussed how your values compare to average/normal values for metals in limestones or shells as well as how they compare to values that have been linked with malformations (compare reviewer 4). The currently available data suggest that your reported values are quite like normal (background) values of heavy metals in such settings. This is further supported by the fact that the increase of heavy metals is closely linked to tuffite samples (BL-5 to BL-7; compare reviewer 4).

2. Metal toxicity as driver: I greatly appreciate you sampling of brachiopod shells and more equal discussion of alternative hypotheses, but the results are not conclusive enough to firmly conclude metal poisoning as the main cause of the malformations (see reviewer 2 and reviewer 4). In this respect, you have two options:

a. If you want to be able to have firm support for the metal poisoning, you need to make sure that the shell is robustly sampled by micro-drilling or LA-ICP-MS of section shells and screening for diagenetic alteration is performed (compare reviewer 4) and appropriately documented in paper or its supplementary material (grinding up shells is insufficient for these purposes as their preservation status needs to be properly evaluated and its needs to be avoid that sedimentary infill or internal sediments is sampled with it – also there is considerable overlap between deformed and non-deformed samples). More importantly, this conclusion of metal poisoning can only be firmly made if the results of these analyses show that non-diagenetically altered shells of brachiopods with deformations have significantly higher concentrations for heavy metals than non-diagenetic altered shells of similar brachiopods from beds/rocks before and after which are not deformed, and where the source rocks have similar (background) values (compare reviewer 4). This option would require major revisions as it involves new analyses and re-evaluations.

b. Alternatively, the potential influence of metal poisoning and the relevant data can still be presented and discussed (reviewer 2) but it should be more clearly concluded that with the current data/analyses no firm evidence is available that metal poisoning was responsible for the observed malformations (see also point of high background concentrations in source rock and tuffites). If you choose this option, I would like to see explicitly mentioned how the hypothesis could be further tested in future studies (e.g., through the type of analyses mentioned under option 1). This option would align with a minor revision decision.

3. Formatting/Typographical issues: some typographical and style issues remain including the headings in supplementary file 1 (see reviewer 3)

I greatly apologise for the delay in making my decision but obtaining reviews and appropriately evaluating the divergent recommendations proved more time-consuming than anticipated. Please address the main points listed above as well as additional points raised by the reviewers including those in annotated pdfs. I reiterate my point to balance the conclusions to the used methods/ analyses and results. In case of point 2 – doing additional analyses (option 2a) would align with a major revision decision while option 2b would rather align with a minor revision decision.

I look forward to receiving your revised manuscript whatever option you decide.

Reviewer 1 ·

Basic reporting

This is well written and interesting paper on the malformations in Late Devonian brachiopods. The authors have improved the MS. Now it can be published in its present form.

Experimental design

Research question are well defined, relevant & meaningful. Rigorous investigation have been performed to a high technical standard. Methods are described with sufficient detail.

Validity of the findings

All underlying data have been provided; they are robust, statistically sound. Conclusions are well stated, linked to original research question and are supporting results.

Additional comments

Can be published in its present form.

Reviewer 2 ·

Basic reporting

I see that the authors made substantial improvements to the previous text. Although they still prefer the toxic metal poisoning as a possible driver of brachiopod shell malformations, they do not entirely 'escape' from epi/endoparasitic factors. I think that the toxic metal, especially Pb, influence is possible and interesting hypothesis and may be presented as such, but still the data is not as supportive as one should wish them to be. The higher content of Pb, for example, is only significantly higher in one brachiopod species, while in the other the its content, as the contents of other metals, is rather similar or insignificantly higher. The same concerns the Cd content in the horizon with malformed shells and the horizon below: they are very similar and not significantly different.
Anyway, the conclusions are balanced and the hypothesys possible, at least concerning the influence of Pb, so I think it is worth to show the results. Thus, I recommend the publication, let it be read and discussed.

Experimental design

OK. I don't see any flaws here,

Validity of the findings

OK.

Additional comments

No.

·

Basic reporting

Thanks for sending me a revised version of the manuscript! It reads much better now!
I have added only a few remarks (style/typing errors) directly into the PDF file.

One important thing is to change also the headings in <Supplemental file 1.>, in order that these conform with the corrections made in the running text:

normal/abnormal specimens

Please consider the newly added corrections - as soon as that is finished the paper is ready for publication from my side!

All kind regards,
Thomas S.

Experimental design

'no comment'

Validity of the findings

'no comment'

Additional comments

see 1. Basic reporting

·

Basic reporting

The manuscript has been significantly improved by including the discussion of parasites, providing more information on the geological setting, nature of malformations, and toning down the discussion on the influence of metal poisoning.

Experimental design

The new analysis of four brachiopod shells is rather meaningless as grinding up the total shells for analysis would have included the sedimentary infill or internal cement. What is really needed is the careful analysis of shell material. This can be done by micro-drilling the shell, or by LA-ICP-MS of sectioned shells and should be accompanied by a careful evaluation of shell diagenesis before analysis – see Baeza-Carratala et al. 2021 (Estudios Geologicos, 77(2): e141) for an example.

Validity of the findings

I appreciate that the authors analysed some brachiopods to look for support for their interpretation that the malformations are linked to metal poisoning. However, I remain highly unconvinced for the following reasons:

1. What constitutes excessively high metal values in limestone? What values would be considered average / normal for metals such as Pb, Cd, or As that are known to induce tetatomorphies? I had a quick look and found two studies on similar rocks providing relevant data: Franchi et al (2015, Chemical Geology, 409: 69-86) describe the elemental composition of Devonian limestone from seamounds and associated sediments, and Babek et al. (2021, Sedimentary Geology, 420: 105934) describe the elemental composition of Ordovician orthoceratide limestone (see tables below). Furthermore, Baeza-Carratala et al 2021 (Estudios Geologicos, 77(2): e141) analysed many individual brachiopods shells from the Lower Jurassic and their values also suggest that the metals of the current study are quite average. Clearly the current study has very similar values for the relevant elements as these studies (see attached table), which raises the question if this is not just the normal background amount of heavy metals to be expected in such settings.
2. The increase of heavy metals described for the section is closely linked to samples BL-5, BL-6, and BL-7, all of which are from tuffites (all other samples are from bioclastic limestone). It is not a big surprise to get higher values of heavy metals in these sediments.
3. See experimental design above

Additional comments

The way forward to provide a robust analysis of whether the malformations were induced by metal poisoning is to carry out is to focus on the brachiopod shells and analyse them with the methods and care outline above. And this should not just include the shells from the layer with the malformations, but older/younger layers with similar brachiopods as well. I would only be convinced if the shells from this particular horizon with the malformations contains contain significantly higher metal concentrations than other horizons that lack malformations. In the absence of such evidence, I would strongly recommend to remove any statements that suggest that the malformations are linked to metal poisoning. The topic of metal poisoning can still be discussed and the relevant data presented. But rather, the discussion should conclude that there is no evidence that metal poisoning was responsible for the observed malformations.

---

## Round 0.3 · Minor Revisions

Thank you for thoughtfully addressing our suggestions. I apologise for the delay in getting back to you, but it proved difficult to find sufficient reviewers to re-review. For this reason, I also did an additional (in-depth) review. I feel the manuscript has greatly improved and close to acceptance, there are just some additional typographical/language/formatting issues as well as some clarifications concerning previous (paleo)pathology work and geochemical work which I would like you to address (see particularly comments by reviewer 2) before publication. Please address all points raised by the reviewers and me including those in annotated pdfs.

Thank you for your patience and I look forward to receiving the revised manuscript and finally being able to see this research published.

Reviewer 1 ·

Basic reporting

Clear and unambiguous, professional English is used throughout. Literature references, sufficient field background/context are provided. Authors interpretations are supported by the data.

Experimental design

Original primary research within Aims and Scope of the journal. Research question are well defined, relevant & meaningful. It is stated how research fills an identified knowledge gap.

Validity of the findings

All underlying data have been provided. The data are fine. Conclusions are well stated.

Additional comments

This paper is well written and deserves to be published in its present form.

·

Basic reporting

Geberally ok. I think, that in the chapter Geological settings should be use and cited papers: Shen et al. (2021- Review of Palaeobotany and Palynology 295) and Stachacz et al. (2021-Acta Geologica Sinica).

Experimental design

no comment

Validity of the findings

no comment

Additional comments

Dear Editor,

I was reading with real satisfaction this manuscript by Ruiwen Zong and Yiming Gong entitled Malformations in Late Devonian brachiopods from the western Junggar, NW China and their potential causes. In the present manuscript, Ruiwen Zong and Yiming Gong did nice work to explore the Late Devonian brachiopods shell malformation caused by epi/endoparasites infestation. An interesting topic and this paper will be quite good for PEERJ. I recommend it can be published after minor revisions after adding the comments I proposed below. The main issues I am concerned are:

1. In my opinion, these malformations are connected rather with biotic interactions with parasites. Oxygen depletion is not a reliable factor in the investigated section in the Hongguleleng Formation. We can exclude anoxic conditions. The U/Th ratio can be confusing especially in pure limestones, and higher values of these ratios are often connected with impoverished with detrital fractions represented by Th. But it did not reflect true redox conditions. This ratio was proposed by Jones and manning (1994) to investigate ancient mudstones rocks in fact. Generally, it is very important to look at directly U contents and enrichment factors. In fact, the better way to decipher redox changes is trace element enrichment factors (EFs). Therefore caution and controlling it by other proxies are needed when we use simple ratios such as U/Th, V/Cr for detail see Algeo & Liu, 2020(Chem. Geol. vol. 540) and Rakociński et al., 2021(3-Paleo vol. 566).

2. In the investigated sections U contents are very low < 1 ppm in beds with malformed brachiopods (samples BL-3 and BL-4). UEF in BL-3 and BL-4 reached 0.67 and 0.18 (in comparison to average limestones according to Wedepohl, 1970). U is depleted which is indicative of oxic conditions in fact. It is confirmed by very low Mo contents < 1 ppm in the whole section. According to Suttner et al. (2014), the TOC contents in the whole Honggulelung Fm is very low with max values of 0.22 wt. %. Generally close to 0.1 wt. % TOC. These sediments are organic poor, and they consist of a lot of benthic biotas which is indicative of oxic conditions. Additionally, Stachacz et al. (2021-Acta Geologica Sinica) described rich ichnofossils in the lower part of Honggulelung Fm which reflected not oxygen restricted benthic environment.

3. High levels of heavy metals or toxic elements can also lead to malformation of organisms. Yes, that is true, but it is rather not in the investigated case. Authors stated that “The levels of some heavy metals (e.g., cadmium, lead, barium, and zinc) in the layer with the malformed athyrid shells are relatively high compared to the levels in the lower part of the Lower Member, particularly cadmium and lead.”
However, values Cd and Pb are lower in samples BL-3 and BL-4 than in the upper part of sections (see Supplementary files). On the other hand, Pb can be enriched in sediments during diagenesis. Additionally, Pb values 5 ppm are not high in fact. Cd values also are not too high, I have higher values e.g. in pure fossils-rich limestones (crepida Zone at Kadzielnia in the Holy Cross Mountains) reached values between 0.1 to 4.1 ppm. Additionally, barium, lead, and zinc can be enriched during diagenesis by hydrothermal fluid migration, but in the investigated section they are not very enriched in fact. In summary, I think the best way to explain these brachiopod malformations by authors is biotic interactions.

4. I know, that it is not a geochemical paper, but I think, that in results you could short describe more important geochemical data, which were used in the discussion?

5. In the chapter Geological settings should be use and cited papers: Shen et al. (2021- Review of Palaeobotany and Palynology 295) and Stachacz et al. (2021-Acta Geologica Sinica).

Other minor remarks:
Line 32: cephalopods: could cited e.g. House, 1960-Palaeontology, Klug, 2007-Acta Palaeontologica Polonica, De Baets et al., 2011-Acta Palaeontologica Polonica; especially I recommend cite here De Baets et al. 2015-Ammonoid Paleobiology.

Lines 74-79: That is not true, we know many successions rich in fossils from Lower Famennian. Reef organisms are rare generally to Middle Famennian, but other groups are numerous and diverse in fact. What is important according to Shen et al. (2021-Review of Palaeobotany and Palynology vol. 295) Hongguleleng Formation starts from crepida conodont Zone and is lack triangularis Zone, this is a recovery interval after the Frasnian-Famennian crisis, therefore biodiversity increases.

Lines 199-203: It is worth mentioning that TOC contents in the studied formation are very low (for detail see Suttner et al. 2014).
Lines 252-253: “The lead contents are 1.3 and 2.2 ppm in the lower part, but 5.1 and 5.2 ppm in the upper part.” But in higher samples, the Pb contents increase > 7 ppb.
Lines 254-256: I think that such small variations in trace metals are irrelevant.
Lines 263-268: I’m not sure whether it is a statistically significant sample. However, it could be interesting.
Lines 272-273: Good point. I agree with this conclusion.
Lines 283-290: Very good point and future challenges!
I think, that in the future may be a good way will be cutting malformed shells to try to find parasites (see De Bates at al. 2011-Acta Palaeontologica Polonica).

Kind regards,
Michał Rakociński
University of Silesia in Katowice
Faculty of Natural Sciences
Będzińska 60
41-200 Sosnowiec, Poland
E-mail: michal.rakocinski@us.edu.pl

---

## Round 0.4 · accepted · Accept

Thank you for addressing our suggestions. I just would like to make sure to avoid the terms deformation and deformed in the wrong context to avoid confusion in the final manuscript. I advise you to consistently use malformation or malformed as well as abnormalities in a more general context (see annotated pdf). Please also replace rate by ratio which would be more appropriate.